

# Improved Arctic Melt Pond Fraction Estimation Using Sentinel-2 Imagery

Kavya Sivaraj, Kurt C. Solander, Charles J. Abolt, and Elizabeth C. Hunke

Los Alamos National Laboratory, Los Alamos, New Mexico, USA

**Correspondence:** Kavya Sivaraj (kavyasivaraj@gmail.com)

**Abstract.** Melt ponds play a vital role in determining the Arctic energy budget by accelerating the rate of sea ice loss aided by their lower albedo. Therefore, an accurate long-term estimate of Arctic Melt Pond Fraction (MPF) is necessary to forecast summer Arctic ice-free conditions. Earth Observation (EO) satellite systems provide ideal tools to monitor the evolution of melt ponds, both spatially and temporally, in near-real time. However, the MPF estimates from these studies are affected

by the presence of small, fragmented ice floes called brash ice, and submerged ice. An improved workflow is necessary to remove the effects of the aforementioned sea ice features from the MPF estimate. Here, we estimate MPF using Sentinel-2 imagery, by coupling a Random Forest (RF) model with mathematical morphological algorithms – morphological dilation and morphological reconstruction – which improves the estimate of MPF by reducing misclassifications from nilas, submerged, and brash ice. Further, we present an inter-seasonal MPF time-series from 2018 to 2021 and show that employing morphological

operations after the RF reduces the mean MPF by greater than 40%. Our results show that the MPF exhibited considerable intra- and inter-seasonal variations, with the maximum MPF reaching as high as 57%.

## 1 Introduction

In recent decades, climate change has led to widespread declines in Arctic sea ice. The Arctic is warming four times faster than the other regions on Earth (Chylek et al., 2022), losing its sea ice extent at a rate of 13% per decade between 1979 and 2020

(Perovich et al., 2002b). The rapid loss of Arctic sea ice has severe implications for the environment, indigenous populations, and biodiversity. A recent study has found that the loss of sea ice could slow down the Atlantic Meridional Overturning Circulation (AMOC) over several decades, which in turn might alter the energy budget of the northern hemisphere (Liu and Fedorov, 2019). More immediately, sea ice loss accelerates the rate of coastal erosion (Barnhart et al., 2014), which might impact the safety of the local population. Furthermore, sea ice decline may open up new navigation routes and areas for natural

resource extraction (Smith and Stephenson, 2013), with negative implications for the Arctic ecosystems.

Melt ponds are a prominent feature of Arctic sea ice, demarcating regions of rapid melting. Melt ponds can be defined as shallow, seasonal water bodies that are formed by the accumulation of meltwater in low-lying depressions of sea ice, which begin forming during the Boreal spring and last into the late summer. As the melt season progresses, the ponds expand in diameter and depth, and they coalesce, forming a series of complex, interconnected structures. The Melt Pond Fraction (MPF)

is defined as



$$MPF = A_{mp}/(A_{si} + A_{bi} + A_{mp}) \tag{1}$$

where $A_{mp}$ is the area of melt ponds, $A_{si}$ is the area of snow-covered ice, and $A_{bi}$ is the area of bare ice, which can reach as high as 53% on the first-year ice during the height of the melt season (Webster et al., 2015). The presence of these ponds lowers the overall albedo of sea ice by 50% (Perovich et al., 2002b) and accelerates sea ice loss by creating a positive feedback between the formation of melt ponds and further melting. Enhanced ice melting increases the exposure of the underlying ocean to the atmosphere, thereby altering the overall energy budget of the Arctic.

The Los Alamos sea ice model, CICE, which functions as the sea ice modeling component of the National Center for Atmospheric Research (NCAR) Community Earth System Model (CESM) and various operational forecasting models (Hunke et al., 2013), already incorporates melt pond dynamics. However, field observations show that these models overestimate the maximum MPF by about 30% (Webster et al., 2022), which could produce biased simulations of sea ice loss. Observation-derived estimates of the MPF are needed to better parameterize these features in models.

Several prior studies have attempted to characterize and quantify melt ponds using optical and microwave satellite imagery. Markus et al. (2002) used true color images obtained from 30 m resolution Landsat 7 and established that melt ponds can be differentiated from the surrounding sea ice features using the spectral differences of bandwidths. Tschudi et al. (2008) used surface reflectance data obtained from different MODIS (MODerate resolution Imaging Spectroradiometer) MOD09 (spatial resolution: 500 m) bands to infer a rapid seasonal growth of melt ponds to a maximum MPF of 40%, followed by a slow decline in August. Zege et al. (2015) used data collected from the Medium Resolution Imaging Spectrometer (MERIS) level 1B products with a spatial resolution of 1 km to retrieve the MPF. They employed the iterative Newton-Raphson method to simultaneously obtain the ice and melt pond properties. This allowed them to account for the variability in the pond reflectance brought about by melt progression. Webster et al. (2015) used 1 m resolution panchromatic imageries obtained from National Technical Means (NTM) and estimated that the MPF reached as high as 53% in first-year sea ice over the Beaufort and Chukchi seas. Wang et al. (2020) improved MPF estimates by 30% by employing a technique called LinearPolar algorithm. This algorithm leveraged spectral combinations of blue and near-infrared bands obtained from S-2 and estimated MPF as a linear function of spectral intensity, after transforming it into polar coordinates, by assigning each pixel along a ramp that varies from 100% pure sea ice to 100% pure melt pond. Similarly, Niehaus et al. (2023) developed a formulation to generalize the LinearPolar algorithm and successfully applied it to different sea ice conditions. Xiong and Ren (2023) used MODIS MOD09 products coupled with dynamic endmember reflectance modeling to estimate the MPF between 2000 and 2021 over the pan-Arctic region.

Prior studies of melt pond dynamics are limited by the low spatial resolution of satellite images. This is particularly true for the MODIS-based data products which are unable to identify individual melt ponds that measure up to a few hundred meters in diameter. Although a few studies (Webster et al., 2015; Perovich et al., 2002a) utilize finer-scale aerial photographs whose pixel resolution is higher than the Sentinel-2 images used in our study, they were taken as part of aerial field campaigns and offer only limited spatial and temporal coverage. It would thus be impossible to extend these studies to pan-Arctic regions for several



melt seasons. As we will show, the accuracy of MPF estimates from such studies can also be affected by misclassifications
caused by ice floe edges and smaller ice fragments (< 2 m across) present between colliding ice floes, also known as brash ice
(Webster et al., 2015). Similarly, although Niehaus et al. (2023) estimated the MPF for the pan-Arctic region between 2017
and 2021, their generalized LinearPolar algorithm approach does not explicitly account for brash ice. Considering that, as we
will show, brash ice has similar spectral characteristics to melt ponds. Although brash ice might not be a common feature over
the region studied by Niehaus et al. (2023), owing to their study area being dominated by multi-year ice, it is more prone to
be developed over regions dominated by first-year ice which has increased from 35%-50% in March of 1980 to over 70% in
March of 2019 (Perovich et al., 2020). As Arctic sea ice has been trending towards thinner and structurally weaker ice in recent
years (Sumata et al., 2023), brash ice will become more common, increasing uncertainties in future MPF estimates. Therefore,
a workflow is needed to account for these features, to avoid misclassifications and improve the accuracy of MPF to better
parameterize melt ponds in ESMs.

Our study builds upon these previous works by utilizing finer-scale (10 m) satellite imageries obtained over multiple melt
seasons to alleviate the pitfalls associated with employment of coarser resolution imagery obtained over a single season. The
higher spatial resolution improves MPF estimates by mapping individual melt ponds that may have gone undetected in prior
studies. An important aspect of our study is incorporation of morphological image processing techniques, which reduces the
effects of biases from ice floe edges, nilas, and brash ice. Finally, the multi-year satellite observations used in our study provide
one of the few time-series constructions of high-resolution, inter-seasonal MPF over an Arctic domain.

## 2   Study area and data acquisition

Our study area comprises  150 km$^2$ bounded by 72.044°N and 71.934°N latitudes and 127.836°W and 128.217°W longitudes
off the northern coast of Alaska and Canada (Fig. 1). This region is dominated by seasonal ice on average (Tschudi et al.,
2019), typically reaching its maximum coverage in March and minimum in September. This location is populated by variable
brash ice floes, nilas, and submerged ice throughout each melt season, increasing the likelihood of biases in MPF estimates
through conventional melt pond identification methods, making it an ideal location to test our workflow.

To estimate MPF, we use 39 optical images from the European Space Agency's (ESA) Sentinel-2 (S-2) satellite (ESA,
2022a). S-2 has two sensors – Sentinel-2A and Sentinel-2B – which orbit the Earth 180° from each other. S-2 provides true
color optical imageries at a spatial resolution of 10 m and a temporal resolution of less than six days over the polar regions.
We use data from 2018 to 2021 between the months of May and August to capture the melt season. MPF estimates derived
from S-2 were validated against higher-resolution (pixel size less than 2 m), true-color optical images from the WorldView-3
(WV-3) satellite, acquired on June 27, 2020, and July 11, 2022 (Maxar, 2022).



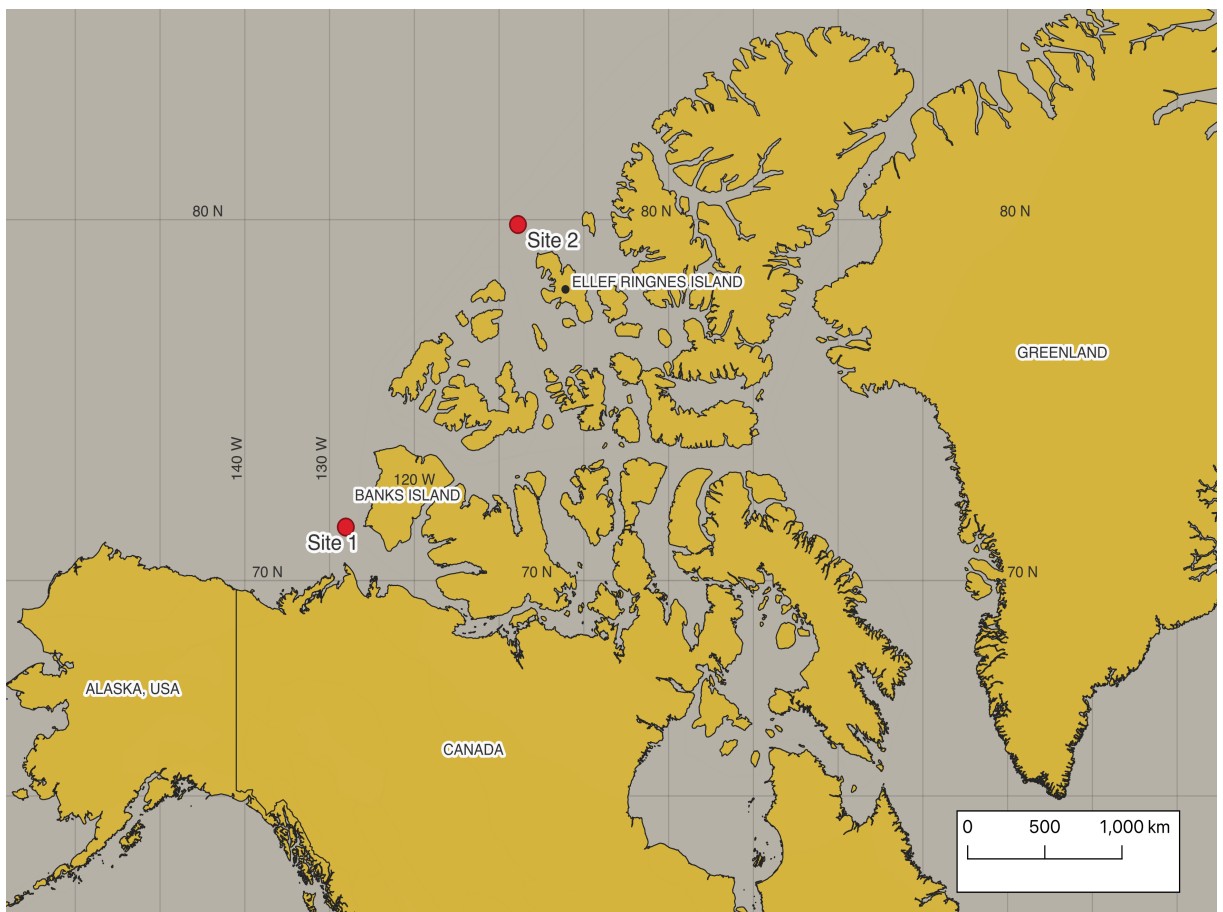

**Figure 1.** Study area, showing sites 1 and 2, which were both used in validation.

## 3  Workflow

A schematic representation of our workflow is presented in Fig.2. Our algorithm consists of five stages, explained in further
detail below.

### 3.1  Stage 1: Data preprocessing

Images were preprocessed to filter poor-quality data caused by atmospheric interference such as cloud coverage and shadows.
The preprocessing steps were carried out in the European Space Agency's (ESA) Sentinel Application Platform (SNAP) 9.0
software (ESA, 2022b). The study area is covered by a mosaic of two S-2 images on a given day, which were stitched together
to facilitate further processing using the SNAP software. To overcome the interferences associated with the S-2 data, only the
level-2 products from the Copernicus Hub of ESA were used for the years 2019, 2020, and 2021. Since level-2 products were



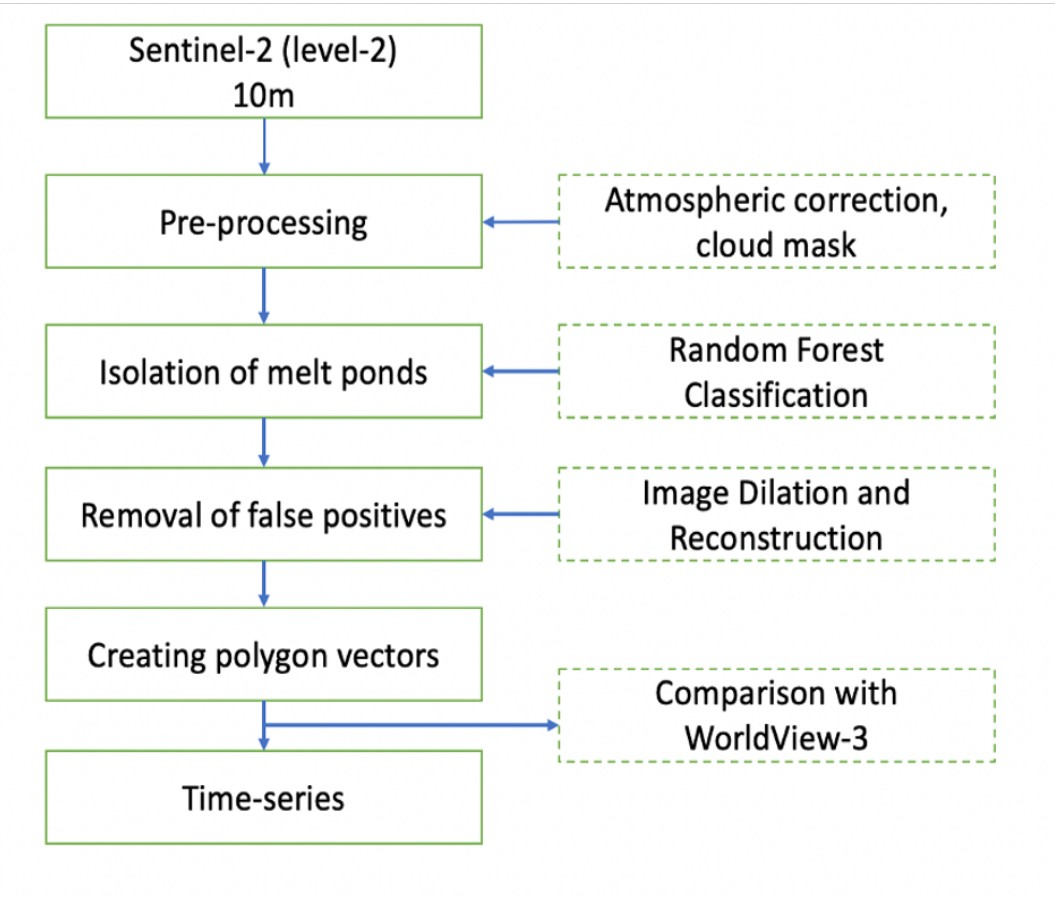

**Figure 2.** S-2 data processing workflow to generate MPF time series

not readily available for images taken before 2019, level-1 products were downloaded for 2018 and atmospherically corrected using ESA's Sen2cor software version 2.5 (Telespazio VEGA, 2018). The atmospheric correction accounts for thick, opaque clouds; pixels covered with thin cirrus clouds were excluded from the study using manually generated cloud masks through
visual interpretation.

### 3.2    Stage 2: RF classification of melt ponds

Melt ponds were distinguished from the surrounding sea ice and open water by applying the RF classification in SNAP 9.0, which is a supervised learning algorithm that automatically classifies features by training on user-labeled data. RF uses an ensemble technique called bootstrap bagging to create a user-specified number of subsets from the training data and make the
final classification decision based on majority voting to reduce overfitting (Altman and Krzywinski, 2017). The training pixels were manually selected from two images taken on June 27, 2020, over a much larger area of 18,326 km$^2$, which was chosen



to capture training pixels of as many melt ponds of different sizes and intensities as possible. For the S-2 imagery, a total of 11,588, 13,307, 13,214, and 15,595 training pixels were selected for four identified surface feature classes: melt ponds, bare ice, open water, and snow-covered ice, respectively. The classification was repeated using ensembles of 10, 100, and 300 decision
trees, and 100 was selected as the optimum number in the final analysis based on computational efficiency and performance.

## 3.3    Stage 3: Removal of misclassifications

The next step of the workflow involves the application of image processing techniques to remove false positive misclassifications. One common feature of Arctic sea ice during the melt season is the presence of thin ice layers called nilas. Exceptionally thin ice formations in contact with the underlying seawater tend to mimic the spectral signatures of melt ponds in true-color
images. Additionally, brash ice and edges of sea ice that are in direct contact with the open ocean may exhibit the spectral properties of melt ponds and were frequently mislabeled by the RF classifier as melt ponds. To remove these misclassifications, we applied two morphological image processing techniques – morphological dilation and morphological reconstruction (Soille, 2003) – to all the S-2 images, and their comparative performance was assessed visually.

The morphological dilation algorithm takes binary images as input and stretches their boundaries in all directions by a
specified number of pixels (threshold). Misclassifications caused by submerged and brash ice were commonly sandwiched between the ice floe edges and open water pixels. Thus, the open water pixels were targeted for dilation using the morphological dilation algorithm available in the filtered bands module of SNAP 9.0. This process extended the boundary of open water pixels, and the resultant dilated pixels now included the sandwiched misclassifications. The resultant dilated pixels (misclassifications) were then subtracted from the melt ponds (true melt ponds + misclassifications) obtained through RF classification to isolate
the true melt ponds. Three dilation threshold values ( 3, 5, and 7 pixels) were experimented with, and the optimal dilation threshold was determined visually. A threshold of 5 pixels performed better in removing false-positive melt ponds caused by nilas, submerged ice, and brash ice pixels for images acquired between May and June, and for images acquired between July and August, the optimal threshold was found to be 3 pixels. We found that increasing the threshold further resulted in masking out more true melt ponds that were closer to ice floe edges and a bigger portion of ice floes themselves. However, we note that
the optimal threshold value is site-specific and may vary for different locations and ice conditions.

The morphological reconstruction algorithm, available in Scikit-Learn, was applied to binary images of melt ponds and open water. It takes two binary images as input: a seed image and a mask image. The seed image consisted of melt pond pixels returned by the RF algorithm (true ponds + misclassifications) and open water pixels, whereas the mask images consisted of only open water pixels. The algorithm identified the instances where the melt ponds and open water touched each other and
returned those pixels. Since the true melt ponds formed on the ice floes and were typically not in contact with open water, the pixels returned by the algorithm mostly consisted of sandwiched misclassifications, which were then subtracted from the true melt ponds. Although morphological reconstruction performed better in removing misclassifications produced by nilas, submerged ice, and brash ice, it is more prone to removing true melt ponds occurring close to open water pixels. In contrast, the tendency of morphological dilation to remove the true melt ponds can be limited by adjusting the threshold.



In the final construction of the MPF time-series, either morphological dilation or morphological reconstruction was applied to each image, based on their performance. To assess the performance of the morphological operations, misclassifications were removed manually through visual inspection and the estimates were compared with those obtained after applying the morphological operations. The mean difference in misclassifications from these estimates was less than 1%.

### 3.4     Stage 4: Vectorization and time series generation

The snow-covered ice, bare ice, and melt ponds identified in the previous steps were converted into polygons whose areas can be used to estimate the total MPF, using a Connected Component Analysis (CCA) technique (Shapiro, 1996). The CCA algorithm takes binary images as input and scans them to identify connected pixels, which are then grouped together to form polygons. Two pixels are considered connected if they are adjacent to each other and have similar intensity values. To perform CCA, the features module available in the Rasterio library (Sean Gilles, 2021) was used. The algorithm was applied to the

binary images of each feature class to create their corresponding polygon vectors, which were then used to estimate the MPF of each S-2 image using Eq (1) and construct an MPF time-series.

### 3.5     Stage 5: Validation

To validate the MPF estimates, two S-2 images were compared against coincident WV-3 images acquired over the study region (Site 1) on June 27, 2020, and another region  100 km north of Ellef Ringnes Island on July 11, 2022 (Site 2). Site 1 covered

an area of  150 km$^2$, whereas Site 2 covered  10 km$^2$. These regions were selected for validation due to the minimal difference in the acquisition time between both images, which enables better comparison of melt ponds on the same ice floes. For Site 1, the melt pond estimates were obtained using the morphological dilation algorithm to reduce misclassifications from the WV-3 image, and the morphological reconstruction algorithm for the S-2 image. The same process was carried out for Site 2, but in this case, morphological reconstruction was applied to both the S-2 and WV-3 images because it performed better than dilation

in removing the misclassifications in both of these images. Moreover, in both the WV-3 images, the melt ponds with areas less than the minimum pixel area of S-2 (100 m$^2$) were removed from further analysis since these melt ponds are not discernible in S-2. To perform validation, we compared the histograms of individual melt pond areas obtained from S-2 and WV-3. Since WV-3 images have a higher pixel resolution of less than 2 m, the estimates from WV-3 were taken as a baseline for comparing the estimates from S-2.

## 4     Results and discussion

### 4.1     Misclassifications through melt pond stages

Figure 3 shows the estimated MPF for each melt season between May and August from 2018 to 2021. The blue dots (unadjusted) represent MPF values obtained directly from RF classification, and the orange dots (adjusted) represent the MPF values estimated after reducing the misclassifications using either the morphological dilation or morphological reconstruction method



**Table 1.** MPF estimates of coincident S-2 and WV-3 images.

| Image | Date of Acquisition | Unadjusted MPF in % | Adjusted MPF in % | Adjusted MPF (above 100 m$^2$) in % |
|-------|---------------------|---------------------|-------------------|--------------------------------------|
| WV-3 | 27 June 2020 | 8.7 | 8.0(including melt ponds of area less than 100 m$^2$) | 5.0 |
| S-2 | 27 June 2020 | 10.7 | 5.4 | 5.4 |
| WV-3 | 11 July 2022 | 13.3 | 12.0(including melt ponds of area less than 100 m$^2$) | 7.3 |
| S-2 | 11 July 2022 | 9.5 | 7.1 | 7.1 |

followed by manual reduction (Section 4.4). We identified three melt stages for each season based on changes in melt pond characteristics determined visually. In Stage 1 (May to early June), the sea ice surface was dominated by snow cover and free of melt ponds (Fig. S1). In Stage 2 (mid-June to early July), the surfaces had distinct, individual melt ponds (Fig.S2). In Stage 3 (mid-July to August), the individual melt ponds coalesced with each other and formed long, narrow interconnected structures (Fig.S3). The misclassifications were highest in Stage 1, with an average difference of 12.35% between unadjusted and adjusted

MPF (Fig.1). Visual inspection showed that the ice floes were snow-covered with no visible melt ponds. The misclassifications were attributed to the widespread presence of nilas and brash ice during this stage (Fig. S4), whose spectral reflectance was similar to that of the melt ponds. The misclassifications were lower in Stage 2, with an average difference of 7.07%, which were primarily attributed to brash and submerged ice (Fig. S5). The scenes in Stage 3 recorded an average difference of 8.40%. The interconnectedness of melt ponds during this final stage and the relatively coarse pixel resolution of S-2 made it difficult

to differentiate among melt ponds, fully melted-through melt ponds, and highly disintegrated ice. Therefore, we acknowledge the possibility of the influence of disintegrated ice and fully melted-through melt ponds on MPF estimates during the late-melt stage. Additional work is necessary to avoid this influence by leveraging finer-resolution imagery. However, that is beyond the scope of this study.

Furthermore, the incidence of misclassification was location-dependent. For example, the S-2 image acquired on June 27,

2020 experienced a difference in MPF of 5.3% after reducing the misclassifications. Another S-2 image acquired on July 11, 2022 at a different location east of Canadian Arctic Archipelago (CAA), which had a lower proportion of brash ice (determined through visual inspection), experienced a difference in MPF of only 2.4% after reducing the misclassifications. This suggests that the influence of misclassifications may be more pronounced in certain regions than in others, depending on ice characteristics. Further, this highlights the necessity of removing the misclassifications to ensure that the MPF estimates are in close

agreement with the ground truth values for all regions of the Arctic and underscores that new methods to appropriately resolve misclassifications, as we propose in this study, are necessary to reduce misclassifications.

### 4.2    Evolution of MPF and melt pond area

As seen in Fig.3, the MPF exhibits significant intra- and inter-seasonal variations, and the time-series captures the overall trend by which MPF tends to increase gradually in most years, reaching a maximum during the warmest months of summer (July and

August). However, neither the onset of melt pond formation nor maximum MPF were observed in all years, due to the limited availability of cloud-free S-2 images. In 2020 and 2021, melt pond onset occurred close to June 12th with an MPF of 35% and



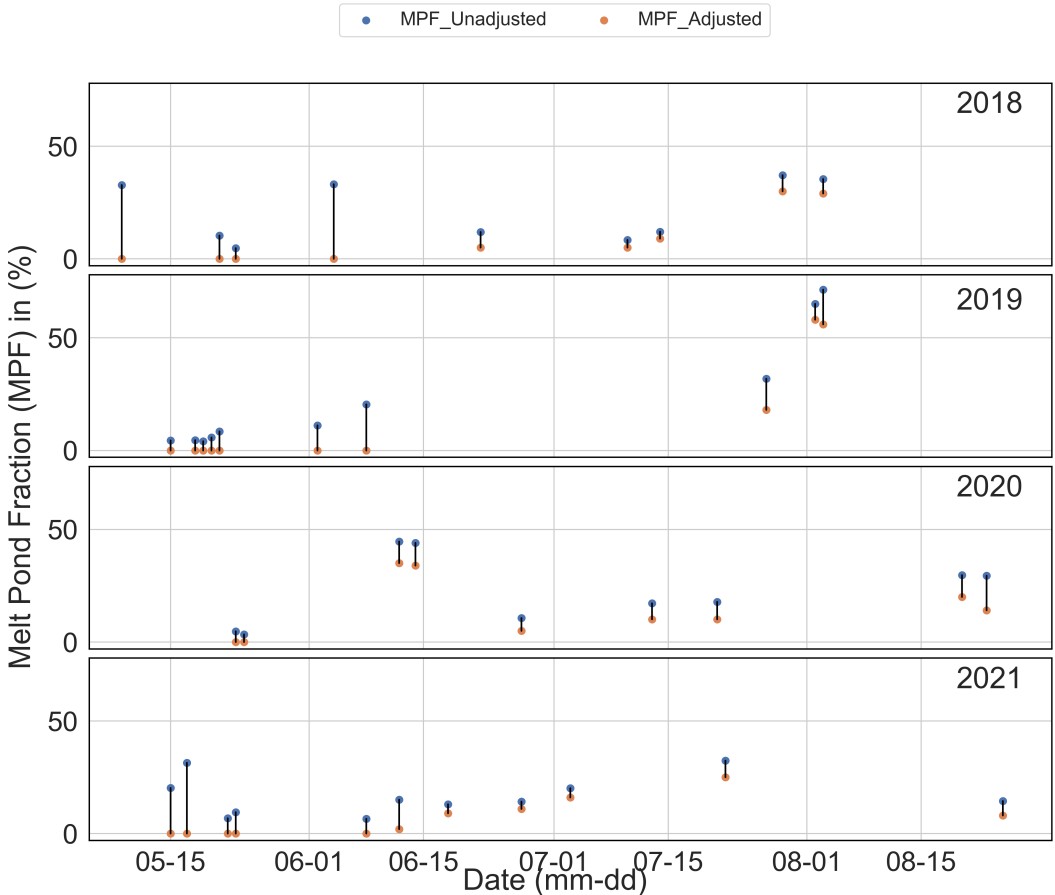

**Figure 3.** MPF through each melt season from 2018 to 2021, as obtained from S-2. The blue and orange dots indicate the unadjusted and adjusted MPF, respectively.

2%, respectively. The timing of melt pond onset was consistent with other studies (Webster et al., 2015; Niehaus et al., 2023). Visual inspection showed the S-2 image on June 12, 2020, had extensive ponding likely caused by flooding, unlike images from other years acquired in mid-June, which could have contributed to the high MPF value (Fig. S6). However, for 2018 and

2019, the onset was not clearly identified due to a lack of sufficient data during June, when melt ponds typically begin forming. The MPF reached 30% and 57% on July 29, 2018 and August 2, 2019, respectively. We do not infer a maximum MPF for 2020 and 2021, as there was a long gap in data record availability between late-July and mid-August when the maximum MPF typically occurs. The large difference between the maximum MPF obtained in 2018 and 2019 could be attributed to differences in atmospheric conditions, ice surface characteristics, and/or composition of ice types. For example, Li et al. (2020) used S-2

and Landsat 8 images acquired over the CAA during the summer of 2017 and found that the first-year ice had a seasonal maximum of 54% and the multi-year ice had a maximum of only 30%. Similarly, Webster et al. (2015) used 1 m resolution





panchromatic images over Chukchi and Beaufort seas and found that first-year ice exhibited a maximum MPF of 53% and multi-year ice exhibited a maximum MPF of 38%. Although ice type composition in our study area was typically dominated by first-year ice (Tschudi et al., 2019), it was populated by a mix of first and multi-year ice floes during late July and early 210 August. Thus, we infer that ice floes sampled during the seasonal maximum MPF of 2018 could be dominated by multi-year floes. Another possibility is the difference in spatial variability of melt ponds for the same ice floe types. For example, Perovich et al. (2002a) reported that the spatial variability of melt ponds differed by an order of magnitude during the 1998 SHEBA field campaign and noted the presence of nearly pond-free first-year ice floes even at the height of the melt season.

We note that the timing and magnitude of maximum MPF obtained from our study do not coincide with Niehaus et al. (2023).
Although their study estimates pan-Arctic MPF covering our study period, Niehaus et al. (2023) do not make a distinction between landfast ice and mobile sea ice in their analysis. Studies show that landfast ice typically exhibits earlier occurrence and higher magnitude of maximum MPF (Landy et al., 2014) compared to other sea ice. Further, the sea ice in the study area analyzed by (Niehaus et al., 2023) is typically dominated by multi-year ice, whereas our study area is located close to the marginal ice zone and is populated by a mix of multi-year and first-year ice late in the summer. Moreover, research suggests
that some multi-year ice in the marginal ice zone might have similar characteristics to that of first-year ice due to flooding and penetration of sea water (Tucker III et al., 1991). Therefore, we note that a direct comparison between our estimates and those of Niehaus et al. (2023) is not feasible due to differences in location and ice type characteristics.

### 4.3 Distribution of melt pond area

Figure 4 shows the area distribution of melt ponds obtained from S-2 images. Unlike the melt pond onset and maximum
MPF, the area distribution of individual melt ponds does not exhibit drastic variations between melt seasons. For example, in 16 of the 21 images with non-zero melt pond fractions, the melt ponds were smaller, exhibiting a median value of 200 m$^2$, suggesting that smaller ponds (< 200 m$^2$) were common across all melt stages and melt seasons. For the remaining images acquired near the peak annual MPF, melt ponds were comparatively larger (median value of 500 m$^2$), due to the coalescence of ponds and drainage channels. Contrastingly, in 2021, larger melt ponds were observed near the middle of the melt season
on June 18, 2021. Interestingly, during 2018, 2020, and 2021, there was a dip in the interquartile range from late June to early July, followed by a gradual increase. The decrease in the interquartile range might be attributed to the drainage of melt ponds, which has been previously reported in several studies (Webster et al., 2022; Perovich et al., 2002a). We cannot rule out the possibility of a similar timing of interquartile range decrease in 2019 due to a lack of available high-quality S-2 images during June and July. However, a similar seasonal decrease in MPF is not evident except in 2020 (Fig.3), despite previous studies
having reported a decrease in MPF due to drainage effects.

### 4.4 Performance of morphological dilation and morphological reconstruction

The performance of morphological dilation varied significantly within and between each melt season. The dilation algorithm was able to remove submerged ice along the ice floes successfully, however, the performance was poor for brash ice and nilas (Fig. 5). The performance was lowest during 2021, with an average difference in MPF of 6.68%, and highest during 2020, with




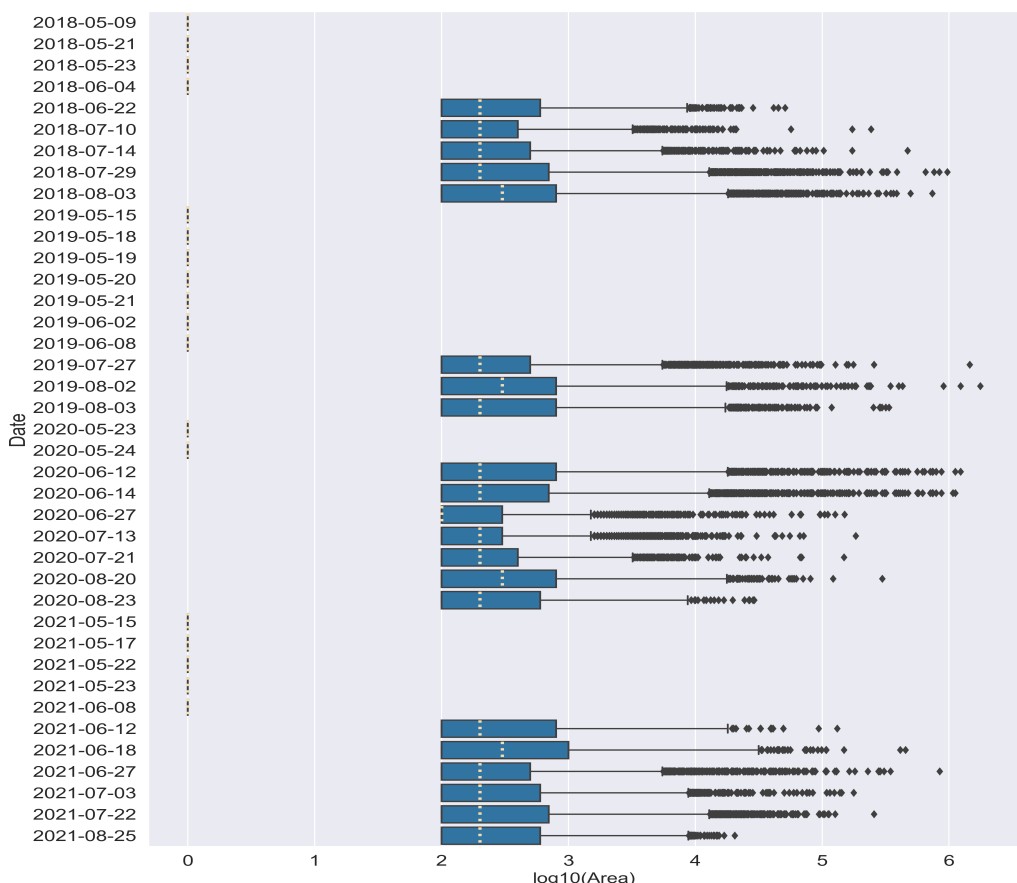

**Figure 4.** shows the distribution of individual melt pond areas in m$^2$ for each S-2 image. The dashed white lines represent the median, the blue boxes represent the interquartile range, and the black diamonds represent the outliers.

an average difference in MPF of 1.11%. Overall, the highest errors occurred in Stage 1, with an average difference in MPF of 6.61%, and the errors were attributable to the widespread presence of nilas. The lowest errors were recorded in Stage 3, with an average difference in MPF of 0.45% (Fig. S7). However, we note that the interconnectedness of melt ponds during this stage made it difficult to differentiate between melt ponds, fully melted-through melt ponds, and highly disintegrated ice, which might contribute to some of the biases in our estimates. In the future, this can be avoided by incorporating finer-resolution imagery.

Similar to morphological dilation, the performance of morphological reconstruction exhibited inter- and intra-seasonal variations. Unlike dilation, reconstruction performed better in removing brash ice and nilas (Fig.5). However, the algorithm exhibited a tendency to remove true melt ponds that were present close to the ice floe edges where the ice pixels were not readily distinguishable. Therefore, this method may not be helpful for regions such as flooded ice where a clear ice-water boundary



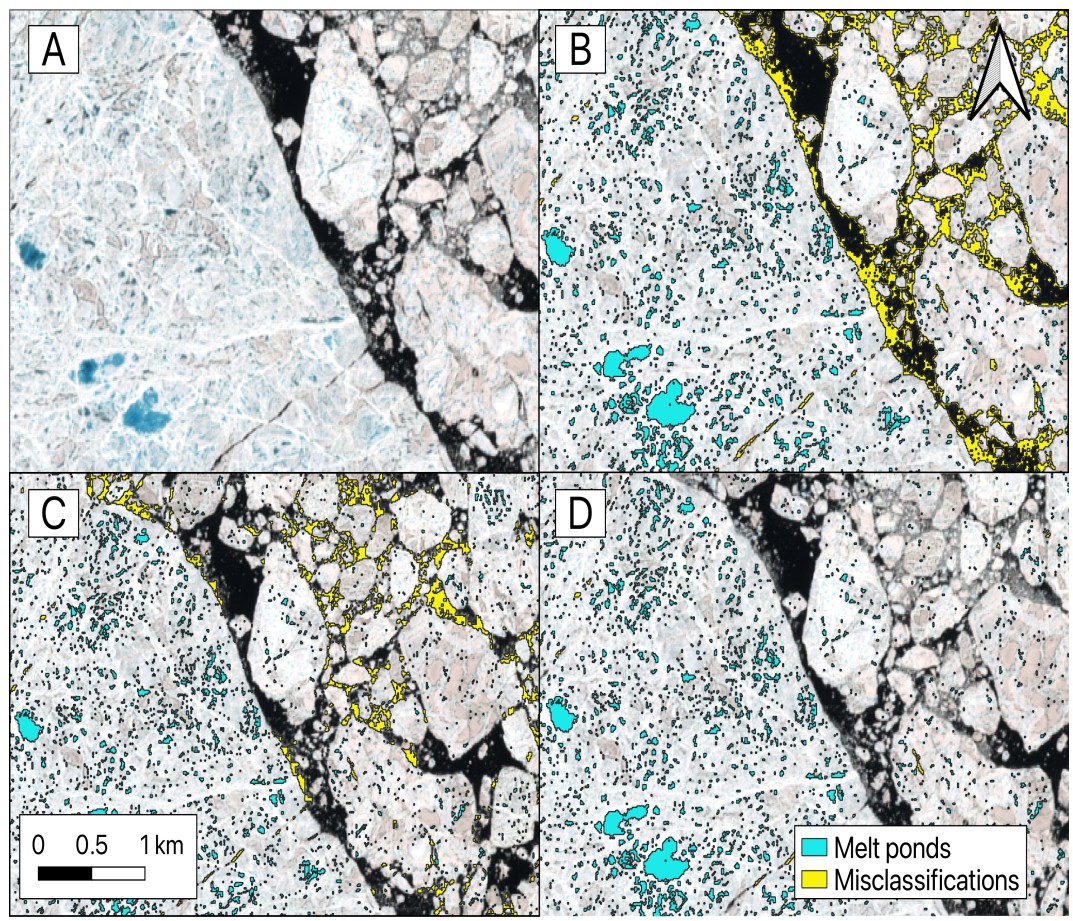

**Figure 5.** A) shows a true color S-2 image acquired over the study area on June 27, 2020, B) shows the true color image overlaid with melt ponds obtained through RF classification. Yellow regions highlight submerged and brash ice misclassified as true melt ponds by the RF algorithm, C) shows reduced misclassifications after the application of morphological dilation, and D) shows reduced misclassifications after the application of morphological reconstruction.

does not exist. The highest performance was recorded in 2021 with an average difference in MPF of 2.45%, and the lowest performance was recorded in 2019 with an average error of 11.73%. In contrast to morphological dilation, the highest errors were recorded in Stage 3 with an average difference in MPF of 13.87%, and the lowest errors were observed in Stage 1 with an average difference in MPF of 1.13% (Fig. S8).

### 4.5 Comparison of S-2 and WV-3

Figure 6 shows a comparison of individual melt pond area distributions between WV-3 and S-2 images. Melt ponds with areas less than 100 $m^2$ constituted 38% and 39% of the total area in the WV-3 images acquired on June 27, 2020, and July 11, 2022,




respectively. However, since these smaller melt ponds are invisible to S-2 due to its relatively coarse pixel resolution, they were excluded from further analysis. For melt ponds larger than 100 m², the performance of S-2 images varied across different size classes. From Fig.6, the 100 m² – 200 m² size class contained the highest number of melt ponds. The S-2 images underestimated the melt ponds less than 500 m² and slightly overestimated those greater than 500 m². The largest underestimation was seen in the 100m² – 200 m² size class, with a difference of 1798 and 1822 melt ponds in Fig.6a and Fig.6b, respectively. Although these two size classes constituted 92% and 89% of the total number of melt ponds identified in WV-3, they represented only 57% and 40% of the total (greater than 100 m²) melt pond area. This discrepancy in the estimates between S-2 and WV-3 is discussed below.

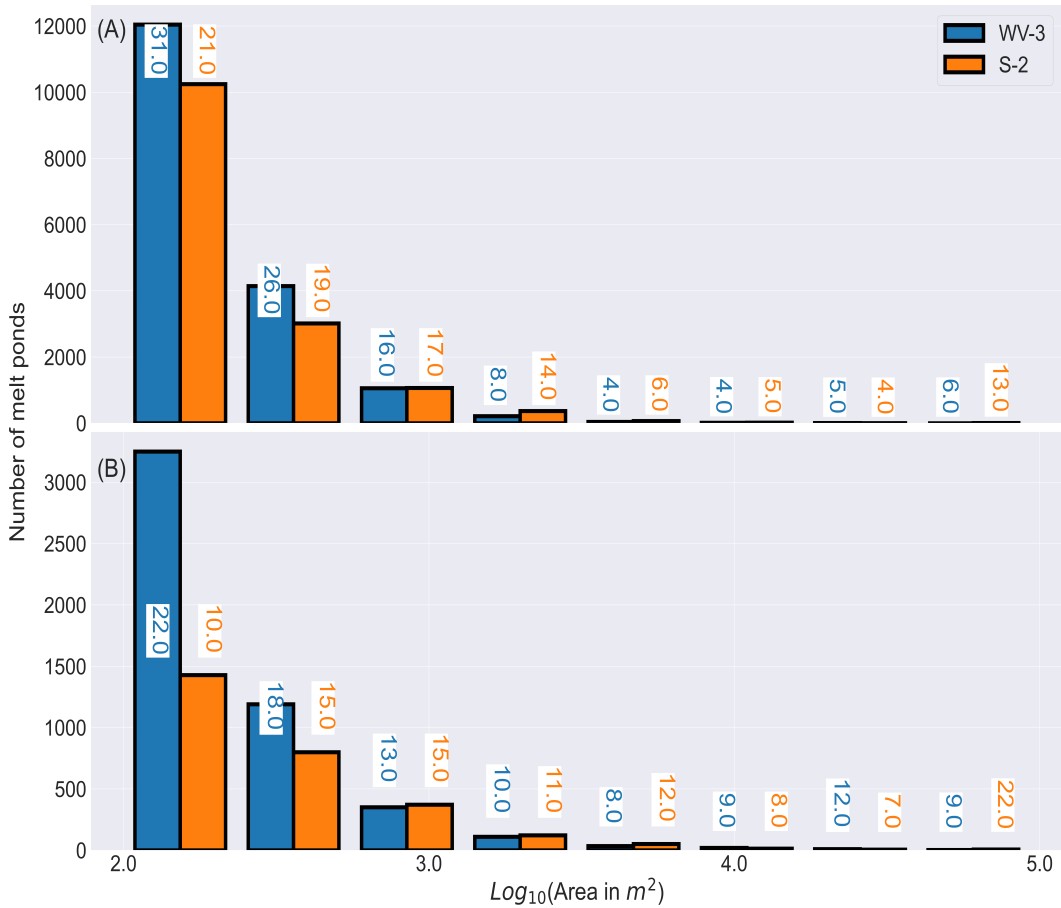

**Figure 6.** Individual melt pond area distributions in coincident S-2 and WV-3 images acquired on A) June 27, 2020, and B) July 11, 2022.

A comparison of coincident WV-3 and S-2 images indicates that estimates from different sensors are affected inequally by the misclassifications. For example, the MPF of WV-3 images acquired on June 27, 2020, and July 11, 2022, experienced a difference in MPF of only 0.7% and 1.3% after reducing the misclassifications, whereas the coincident S-2 images experienced





a difference in MPF of 5.3% and 2.4%. We attribute this difference to the higher pixel resolution of WV-3 (2 m), which is able to correctly resolve smaller brash ice floes into the appropriate ice type, thus resulting in lower misclassification. However, despite considerable variations between the number of melt ponds obtained from S-2 and WV-3 images, the MPF derived from these two data sources showed only minor differences of 0.4% and 0.2% for the images acquired on June 27, 2020, and July 11, 2022, respectively (Table 1, Adjusted MPF above 100 m$^2$). This suggests that the signs of the errors largely balanced out among size classes. We attribute this partially to the incorporation of smaller, closely spaced melt ponds into the higher-size classes in the S-2 imagery, owing to its lower spatial resolution (Fig.S9). Therefore, our results suggest that in spite of the high error for smaller pond sizes, the S-2 images can still be useful in capturing overall MPF variability, as well as the classification of melt ponds of sizes greater than 100 m$^2$.

Although our study captures the changes in MPF through melt seasons successfully, as evidenced by close agreement between S-2 and WV-3 fraction estimates and with other studies, uncertainties in MPF still exist for the following reasons. Our findings from S-2 images show that smaller melt ponds are common across all the melt stages and seasons, as evidenced by the smaller median melt pond area (200 m$^2$). However, the analysis of coincident WV-3 images shows that the melt ponds less than 100 m$^2$ comprise an average of 38.5% of the total melt pond area, suggesting that smaller melt ponds (less than 200 m$^2$) are relatively more common than the larger melt ponds. This is in close agreement with (Buckley et al., 2023), who found that smaller melt ponds (less than 100 m$^2$) can comprise 38% of the total melt pond area by analyzing 18 WV-3 imagery from several locations. Although the similarity between our estimates and that of (Buckley et al., 2023) is encouraging, further research is warranted to determine how consistent the fraction of melt ponds less than 100 m$^2$ is across different locations of the Arctic. Similarly, during melt stage 3, the individual melt ponds coalesced with each other through the connecting drainage channels into narrow interconnected structures, which limited our ability to differentiate melt ponds from frozen ponds, brash ice, and fully melted-through ponds. To account for these shortcomings, we propose that a new augmented dataset might be created by merging the high temporal resolution S-2 and very high spatial resolution WV-3 images.

## 5 Conclusions

We successfully developed a workflow to estimate MPF from S-2 images and showed that misclassifications produced by submerged ice, nilas, and/or brash ice can be reduced by integrating the RF model with mathematical morphological operations to improve the overall accuracy of MPF estimates. Our study found that these misclassifications overestimated MPF by a mean difference of 11% and a maximum difference of 33% between unadjusted and adjusted MPF. Further, our study showed that the misclassifications are location-dependent and might be more important in certain regions than in others, depending on the distribution of brash ice, submerged ice, and nilas. This highlights the importance of removing the misclassifications to accurately estimate the MPF that holds true for all regions of the Arctic. The timing of melt pond onset and maximum MPF are in close agreement with (Webster et al., 2015) and (Perovich et al., 2002a). Further, we showed that the S-2 images can be used to identify overall MPF variability as well as classify melt ponds greater than 100 m$^2$. Despite uncertainties, S-2 imagery remains a valuable tool for estimating MPF because it covers larger parts of the Arctic with shorter repeat times and larger



swath widths. Further, it is publicly available. This study proves that the uncertainties can be reduced successfully using our workflow, which can be applied to pan-Arctic regions, and it is particularly useful for regions of young and structurally weaker ice with a widespread presence of nilas and brash ice. Moreover, our workflow is independent of the spatial resolution of the satellite sensors and, therefore, can be used as a standalone procedure to estimate MPF or as an additional step in other MPF

identification methods if necessary. This underscores the vital role of our study in the efforts to accurately estimate the MPF for pan-Arctic regions. Additional work is necessary to account for melt ponds that are too small ($< 100$ m$^2$) to be detected by S-2, which could be achieved by creating a comprehensive augmented dataset that combines high spatial resolution WV-3 and high temporal resolution S-2.

*Code and data availability.* S-2 images used in the analysis are freely available through European Space Agency's Copernicus Open Access

Hub. WV-3 images are commercially available via Maxar's Global Enhanced GEOINT Delivery (G-EGD) web-based application. All the output files and codes used in this project are available at https://doi.org/10.5281/zenodo.12802216.

*Author contributions.* KCS and CJA conceived the project. KS performed the analysis and interpretation of satellite images with guidance from KCS and CJA. ECH provided guidance on the interpretation of results. KS prepared the manuscript with input and guidance from all the authors. All the authors reviewed the final manuscript.

*Competing interests.* The authors declare that they have no conflict of interest.

*Acknowledgements.* We thank the Center for Space and Earth Science (CSES) of the Los Alamos National Laboratory for partly supporting this work through a research grant. We are also thankful for the support provided, in part, by the Regional and Global Model Analysis Program-funded Interdisciplinary Research for Arctic Coastal Environments (InteRFACE) project, awarded under contract grant 89233218CNA000001 to Triad National Security, LLC ("Triad").



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
