# Peer review of "Improved Arctic Melt Pond Fraction Estimation Using Sentinel-2 Imagery"

_EGUsphere, 2024_

## Referee Comment (RC1)

This study presents an improved methodology for classification of Sentinel-2 imagery for melt pond fraction estimates. Specifically, the combination of random forest models and morphological algorithms allows for the reduction of misclassifications from nilas, submerged ice, and brash ice.

The goal of this study is compelling, and the new methodology presented shows promise. However, the methods section requires greater detail and additional figures to enhance clarity and support the results. I've noted in the specific comments where a supporting figure would be useful. Furthermore, some key details need to be addressed, such as determining the minimum detectable melt pond size using Sentinel-2. Once this is established, a follow-up analysis should be conducted to assess the implications of this threshold on the study's findings.

Specific:

Line 15: how are you citing a 2002 paper for a trend 1979 to 2020. Either reference a dataset, or a more recent paper

Line 29: either a more specific location and time of that study or a less specific number. 50% should be a range or should include "on average", or a specific location.

Line 56: A little misleading to reference Webster 2015 here- because this study utilizes NTM satellite imagery- which has other limitations (coverage, consistency, accuracy) but is not aerial photographs

Line 61: delete "similarly"

Line 62: fragmented sentence starting with "considering that.." is hard to follow

Line 63: long sentence with a lot of thoughts, consider separating into two sentences

Figure 1: consider putting this map in polar stereographic

Figure 1: include a bounding box for the area noted on line 77.

Line 98: does the atmospheric correction mask out areas of thick clouds? What do you mean accounts for them?

Section 3.2: who created the training data, a single sea ice expert, several people, non-experts?

Line 113: what does exceptionally thin mean? I'm sure the WMO nomenclature has specifications for nila definitions.

Line 117: it would be great to have a figure representation of the morphological dilation and reconstruction to eliminate brash ice and nila misclassifications

123: Was this a pixel-based classification then? Figure 5 makes it look like the melt ponds and misclassifications are objects- groups of pixels. Do you eliminate the whole object or just the overlapping pixels? Does that leave some misclassifications in the middle of an area that is misclassified? An example of this would make a great figure

Line 134: how do you create the open water mask that is "only open water pixels"

Line 141: did you assess the performance on all images?

Line 155: why are the two sites such different sizes if you are using a worldview image for comparison at both?

Line 161: it would be interesting to know the statistics of the WV melt ponds that are less than the S2 image pixel size. How many are there? what MPF does this make up?

Line 164: "a higher pixel resolution of less than 2 m" is confusing phrasing

Section 3.5: it would be great to see how the Sentinel-2 and Worldview images look next to one another. Also please note the delta time for acquisition between S2 and WV

Line 170: how much does the MPF change between the dilation/reconstruction methods and the manual reduction? In other words, Figure 3 could include three points indicating the RF output, the morphological reduction, and the morphological +manual reduction.

Line 170-174: how does these stages compare with the Eicken stages of melt pond evolution?

Eicken, H., H. R. Krouse, D. Kadko, and D. K. Perovich, Tracer studies of pathways and rates of meltwater transport through Arctic summer sea ice, *J. Geophys. Res.*, 107(C10), 8046, doi:10.1029/2000JC000583, 2002.

Line 179: show this- show misclassification or ambiguous surfaces listed here- the melt ponds, melted through melt ponds, disintegrated ice.

Line 183: what about other types of misclassifications- melt ponds identified as open water, melt ponds that are not identified (classified as ice), what are other misclassifications that could occur, what is the magnitude of these misclassifications, and in what direction would each of these misclassifications bias the MPF.

Figure 3: include more grid lines.

Line 196: how are you determining melt pond onset? From you MPF estimate? Consider looking at melt onset passive microwave product to compare with your results.

Line 197: if you are going to compare with other studies, make sure you indicate the time and location of the other studies and how you might expect those factors to contribute to the same or different results you are seeing

Line 203: the large difference could be also due to lack of data and not capturing the maximum MPF date

Line 210: populated by mix of ice types in which year?

Line 210: show the image during the seasonal maximum of 2108MPF and see if they are in fact MYI floes.

Line 211: how are you defining "spatial variability"

Line 211: and what is the reason for "spatial variability of melt ponds"- does Perovich 2002 give an answer?

Line 215: is the Neihaus data public? Could you look just specifically at their region to examine the differences?

Line 215: if they only provide Arctic wide estimates of MPF it does not make sense to compare your results with theirs at all- of course the melt timing is different.

Line 225: by "smaller ponds < 200 m^2" doesn't that mean just one single Sentinel-2 pixel? Do you have a limit on the number of pixels required to call the object a pond? This is common of other studies- see Buckley et al 2023 and Perovich et al 2002 papers. If you eliminate small ponds which may be erroneous, you will probably have a totally different distribution to discuss.

Perovich, D. K., W. B. Tucker III, and K. A. Ligett, Aerial observations of the evolution of ice surface conditions during summer, *J. Geophys. Res.*, 107(C10), 8048, doi:10.1029/2000JC000449, 2002.

Buckley, E. M., Farrell, S. L., Herzfeld, U. C., Webster, M. A., Trantow, T., Baney, O. N., ... & Lawson, M. (2023). Observing the evolution of summer melt on multiyear sea ice with ICESat-2 and Sentinel-2. *The Cryosphere*, *17*(9), 3695-3719.

Line 232: see also Buckley et al., 2023 for melt pond evolution for comparison especially because it also uses Sentinel-2 data for MPF.

Figure 4: I think this could be better represented with month and day on the y axis and then the data plotted across the season with each year as a different color- that would be easier to visualize the patterns and how they line up year to year.

Line 239: was the performance the lowest or the presence of errors and misclassifications lowest? How can you know? Are these percentages a percentage of the misclassifications being removed? This is a confusing discussion, please clarify

Line 260: It would be beneficial to establish a lower limit for the detectable melt pond size. Given that a pond in the 100 m² to 200 m² size range corresponds to just a single Sentinel-2 pixel, the discussion around melt pond size distribution within this category lacks robustness. I would be particularly interested in seeing the distribution of ponds starting with a minimum size equivalent to 10 Sentinel-2 pixels. Additionally, it would be valuable to include a dedicated section discussing the limitations of the 10 m resolution, specifically addressing how many ponds and the total pond area may be missed compared to higher-resolution imagery, such as that from WV.

Line 281: There is the stat I was looking for! Maybe include this earlier where noted.

Line 299: The conclusions of the study feel overly optimistic and would benefit from a more balanced discussion of the methodology's limitations. Specifically, the following points should be addressed:

- Limitations of Sentinel-2 coverage: Sentinel-2 imagery is restricted to areas within a certain distance from the coastline, which limits its applicability in more remote regions. Additionally, temporal limitations should be discussed—how frequently are these specific locations sampled, and does this sampling frequency align with the temporal dynamics of melt pond evolution?

- Impact of cloud contamination: A detailed assessment of the extent to which cloud cover renders data unusable would provide valuable context. Quantifying the proportion of affected data could help clarify the reliability of the results.

- Manual aspects of the algorithm: The methodology currently involves manual corrections and decisions, such as selecting the optimal morphological operation. If this approach were scaled up, manual intervention would not be feasible, necessitating a discussion on how these steps could be automated or standardized.

---

## Author Comment (AC1)

Response to Reviewer 1

This study presents an improved methodology for classification of Sentinel-2 imagery for melt pond fraction estimates. Specifically, the combination of random forest models and morphological algorithms allows for the reduction of misclassifications from nilas, submerged ice, and brash ice. The goal of this study is compelling, and the new methodology presented shows promise. However, the methods section requires greater detail and additional figures to enhance clarity and support the results. I've noted in the specific comments where a supporting figure would be useful. Furthermore, some key details need to be addressed, such as determining the minimum detectable melt pond size using Sentinel-2. Once this is established, a follow-up analysis should be conducted to assess the implications of this threshold on the study's findings.

We are very thankful for the reviewer's timely and thorough review of our manuscript. We appreciate their constructive feedback, which we are confident will help us improve our manuscript. As suggested, we have established a minimum threshold for melt pond size derived from Sentinel-2 (S2), and we will conduct a follow-up analysis on how this will affect the melt pond size distribution. Please find below our line-specific response. The reviewer's comments are highlighted in red, and our response numbers are highlighted in blue.

Line-specific response:

Line 15: how are you citing a 2002 paper for a trend 1979 to 2020. Either reference a dataset, or a more recent paper.

Response:1 We thank the reviewer for catching this mistake. We will instead cite the following paper in the revised manuscript:

Perovich, D., Meier, W., Tschudi, M., Hendricks, S., Petty, A. A., Divine, D., Farrell, S., Gerland, S., Haas, C., Kaleschke, L., Pavlova, O., Ricker, R., Tian-Kunze, X., Wood, K., and Webster, M.: Arctic Report Card 2020: Sea Ice, https://doi.org/10.25923/N170-9H57, 2020.

Line 29: either a more specific location and time of that study or a less specific number. 50% should be a range or should include "on average", or a specific location.

Response:2 We will revise the text to "the presence of these ponds lowers the albedo of sea ice significantly. For example, Perovich et al. (2002) noted that the average summer albedo of the multiyear ice experienced a reduction by an average of 50% between April and late July during the 1998 Surface Heat Budget of the Arctic Ocean (SHEBA) field experiment, which started in the Beaufort Sea."

Line 61: delete "similarly"

Response:3 We will delete it in the revised manuscript.

Line 62: fragmented sentence starting with "considering that.." is hard to follow

Response:4 We agree with the reviewer that this sentence is confusing. We propose to add this line: "These features are easily misclassified because they have similar spectral properties to melt ponds" in line 61. We will also remove the fragmented sentence "considering that" from line 62.

Line 63: long sentence with a lot of thoughts, consider separating into two sentences

Response:5 Here we state: "*Although brash ice might not be a common feature over the region studied by Niehaus et al. (2023), owing to their study area being dominated by multi-year ice, it is more prone to be developed over regions dominated by first-year ice which has increased from 35%-50% in March of 1980 to over 70% in March of 2019 (Perovich et al., 2020).*"

We propose to revise the above-mentioned sentence to: "Brash ice can be more common in regions dominated by first-year ice. Further, research suggests that first-year ice has increased from 35-50% in March 1980 to over 70% in March 2019 (Perovich et al., 2020)."

Figure 1: consider putting this map in polar stereographic
Figure 1: include a bounding box for the area noted on line 77.

Response:6 we will add this figure to the revised manuscript.

[Figure]

*Figure 1shows the study area and validation sites. The inset shows the bounding box of the study area. Site 1 = study area and sites 2-5 = validation sites.*

Line 98: does the atmospheric correction mask out areas of thick clouds? What do you mean

accounts for them?

Response:7 The corrected L2A products have an automatic cloud mask, which works well to mask out thick, opaque clouds. However, it could not identify thin, wispy clouds and shadows well. Therefore, these interferences were removed by manually generated masks. We will clarify these points in the revised manuscript.

Section 3.2: who created the training data, a single sea ice expert, several people, non-experts?

Response:8 The training data was created by a single expert involved in the project. The data is available at https://doi.org/10.5281/zenodo.15000902

Line 113: what does exceptionally thin mean? I'm sure the WMO nomenclature has specifications for nila definitions.

Response:9 We will replace this sentence with the WMO sea ice nomenclature and define nila as "a thin crust of sea ice that measures less than 10 cm in thickness."

WMO Sea-Ice Nomenclature: https://library.wmo.int/records/item/41953-wmo-sea-ice-nomenclature, last access: 30 January 2025.

Line 117: it would be great to have a figure representation of the morphological dilation and reconstruction to eliminate brash ice and nila misclassifications

Response:10 This is shown in figure 5. We will also add a conceptual diagram explaining the morphological operations in the revised manuscript.

123: Was this a pixel-based classification then? Figure 5 makes it look like the melt ponds and misclassifications are objects- groups of pixels. Do you eliminate the whole object or just the overlapping pixels?

Response:11 Yes, this is a pixel-based classification. The misclassifications caused by submerged ice, brash ice, and nilas were mostly found to be in contact with open water. Therefore, we leveraged open water pixels to mask out the misclassifications.

The dilation operation extends the open water boundary by a specified threshold into the misclassified pixels that are in immediate contact. This extended boundary is then used to eliminate the overlapping pixels.

After applying the morphological operations, the melt pond pixels and the misclassifications (that remain) were grouped as polygons using Connected Component Analysis (Section 3.4). This feature was used to estimate the area of individual melt ponds (which may be represented by several pixels) and to facilitate the manual reduction of remaining misclassifications. The objects in Figure 5 show these polygons overlaid on the S-2 imagery. These points will be clarified in the manuscript.

Does that leave some misclassifications in the middle of an area that is misclassified? An example of this would make a great figure

Response:12 That is a great question! Yes, these procedures do leave out misclassifications. For example, the extent of brash ice might be higher in some locations than others. Therefore, although a single threshold may be optimized to remove all misclassifications in one area, this may not be the case in another area where misclassifications will remain even after this technique is applied. This is shown in Fig. 5C, where some misclassifications (yellow) still remain after morphological dilation. This can be avoided, to some extent, by adjusting the threshold. However, a higher threshold might mask out a portion of ice floes, thereby removing more "false positives" and still affecting the accuracy of MPF estimation (lines 128-130). The optimal threshold was chosen to be one that did not remove a portion of ice floes while still removing considerable misclassifications. This was determined visually. We will show this in the conceptual diagram.

Line 134: how do you create the open water mask that is "only open water pixels"

Response:13 The reconstruction algorithm takes two binary images as input:
1. A binary raster consisting of melt pond and open water pixels (obtained from RF classification).
2. A binary raster consisting of open water pixels (obtained from RF classification).

We realize now that lines 131-134 might be confusing to a reader, so we propose to rephrase them to reflect the points presented above in the revised manuscript.

Line 141: did you assess the performance on all images?

Response:14 Yes, the performance was assessed for the 39 images used to construct time-series. We clarify this point in the manuscript.

Line 155: why are the two sites such different sizes if you are using a worldview image for comparison at both?

Response:15 The validation site 2 is located north of the Canadian Archipelago, dominated by multi-year ice. This location had features that might be attributed to algae. To exclude these features and minimize their influence in our classification, we used a smaller subset.

[Figure]

*Figure A shows algal-like features*

Line 161: it would be interesting to know the statistics of the WV melt ponds that are less than the S2 image pixel size. How many are there? what MPF does this make up?

Response:16 As the reviewer pointed out below, this information is added in line 281. Please see response 22.

Line 164: "a higher pixel resolution of less than 2 m" is confusing phrasing

Response:17 We will rephrase it to "a higher resolution (1.24 m)" in the revised manuscript.

Section 3.5: it would be great to see how the Sentinel-2 and Worldview images look next to one another. Also please note the delta time for acquisition between S2 and WV

Response:18 We will add a two-panel figure consisting of S-2 and WV-3 imagery, along with the delta time for acquisition, in the revised manuscript.

Line 170: how much does the MPF change between the dilation/reconstruction methods and the manual reduction? In other words, Figure 3 could include three points indicating the RF output, the morphological reduction, and the morphological +manual reduction.

Response:19 We will add this modified Figure 3 to include MPF values from morphological reduction in addition to the RF output and morphological + manual reduction in the revised manuscript.

[Figure]

*Figure 3 MPF through each melt season from 2018 to 2021, as obtained from S-2. The blue, orange, and green dots indicate the unadjusted MPF (RF output), MPF after morphological reduction, and adjusted MPF (MPF after morphological and manual reduction), respectively.*

Line 170-174: how does these stages compare with the Eicken stages of melt pond evolution?

Response:20 Eicken et al. (2002) describe four stages of melt pond (hereafter referred to as "Eicken stage") evolution observed during their 1998 Surface Heat Budget of the Arctic Ocean (SHEBA) field experiment based on the changes in pond coverage. The melt stages identified in our study are based on the physical changes in melt ponds determined visually (e.g., prevalence of distinct, individual melt ponds, vs coalesced melt ponds connected through numerous drainage channels).

- Eicken stage 1 is marked by the onset of melt pond formation with a rapid increase in MPF to sub-maximum. During Eicken stage 1, the ice permeability was very low, allowing the melt ponds on level first-year ice to spread over a larger ice area.

- o The pond onset observed during the Eicken stage 1 was observed in the stage 2 described in our study. The pond onset occurred close to mid-June for 2021, and possibly 2020
- During Eicken stage 2, the MPF decreased, and the ponds shrunk in size due to the drainage of meltwater through cracks and flaws.

  - o . This decrease in MPF was only observed in our 2020 data.

- During Eicken stage 3, the MPF increases significantly to reach the absolute maximum. This stage is also marked by a significant increase in ice permeability and hydraulic head, as well as enlarged flaws that contribute to the disintegration of ice.

- During Eicken stage 4, the ponded areas freeze over, contributing to a decrease in MPF.

  - o These two final Eicken stages were also observed during stage 3 identified in our study. In stage 3, the MPF increased to a maximum in late-July and early-August, followed by a reduction. Here, the melt ponds coalesced into one another through drainage channels, forming a complex, interconnected geometry. However, we cannot say when the MPF reached its maximum in the years 2020 and 2021 due to the unavailability of data.

Eicken, H., H. R. Krouse, D. Kadko, and D. K. Perovich, Tracer studies of pathways and rates of meltwater transport through Arctic summer sea ice, J. Geophys. Res., 107(C10), 8046, doi:10.1029/2000JC000583, 2002.

Line 179: show this- show misclassification or ambiguous surfaces listed here- the melt ponds, melted through melt ponds, disintegrated ice.

Response:21 We will include these as supplemental figures in the revised manuscript.

Line 183: what about other types of misclassifications- melt ponds identified as open water, melt ponds that are not identified (classified as ice), what are other misclassifications that could occur, what is the magnitude of these misclassifications, and in what direction would each of these misclassifications bias the MPF.

Response:22 We thank the reviewer for pointing out these misclassifications. Some of the darker melt ponds can be misclassified as open water. However, in our study area, we found that the darker melt ponds were more common during stage 3 of melting. During this stage (shown in fig. S3), we noticed that the ponds became interconnected with one another, forming narrow, complex structures. Since these structures are too narrow to be identified as distinct melt channels by S2, the whole sea ice area (ponded regions + unponded regions sandwiched between the ponded ones) gets classified as melt ponds (Fig B). Therefore, this might lead to overestimation. We will add this to the revised manuscript.

[Figure]

*Figure B1 shows S-2 imagery acquired on July 16, 2022, with interconnected melt ponds that appear darkened. Figure B2 shows the corresponding RF classification with melt ponds highlighted in blue.*

Since the resolution of S2 imagery is 10 m, melt ponds that are less than 100 m2 can be routinely misclassified as ice. Our analysis from the coincident WV3 image acquired on June 27, 2020, showed that the melt ponds less than 100 m2 constituted 38% of the total melt pond area, whereas the image acquired on July 11, 2022, showed that about 39% of total area was covered by melt ponds less than 100 m2. Further, this is in agreement with Buckley et al. (2023), who found that, on average, 38% of the total pond area was covered by using 18 WorldView images. Since S2 imagery covers a larger area of the Arctic and is publicly available, it is a valuable tool to quantify the evolution of melt ponds. However, to address the limitations of pixel resolution, it is important to estimate the proportion of smaller melt ponds (<100 m2) for different sea ice locations and melt stages to understand their influence on the total melt pond fraction. However, that is beyond the scope of this study, and we hope to address that in future work.

Another misclassification can arise from blue ice, which might be classified as melt pond due to similar spectral reflectance. These blue ice typically occur in glaciers and calved icebergs, forming when the snow layers are compressed under the weight of overlying snowfall. These are typically found near Greenland and are relatively rare in Arctic sea ice.

We realize now that the manuscript might benefit from a dedicated section that talks about the misclassifications mentioned above and others that might occur. Therefore, we will add a limitations section in the revised manuscript in which we discuss the above-mentioned points. However, quantifying these other misclassifications is beyond the scope of this study and can be addressed in future work.

Figure 3: include more grid lines.

Response:23 Please see response 19.

Line 196: how are you determining melt pond onset? From you MPF estimate? Consider looking at melt onset passive microwave product to compare with your results.

Response:24 Yes, we determine the pond onset from our MPF estimate. We thank the reviewer for suggesting using microwave products to compare with our results. We will use the melt onset data from Scanning Multichannel Microwave Radiometer (SMMR), Special Sensor Microwave/Imager (SSM/I), and the Special Sensor Microwave Imager/Sounder (SSMIS) brightness temperature measurements, available through the National Snow and Ice Data Center (NSIDC) to make the comparison.

Snow Melt Onset Over Arctic Sea Ice from SMMR and SSM/I-SSMIS Brightness Temperatures, Version 5: https://nsidc.org/data/nsidc-0105/versions/5, last access: 11 February 2025.

Line 197: if you are going to compare with other studies, make sure you indicate the time and location of the other studies and how you might expect those factors to contribute to the same or different results you are seeing

Response:25 We will include the following information in the revised manuscript.

Here we state: *In 2020 and 2021, melt pond onset occurred close to June 12th with an MPF of 35% and 2%, respectively. The timing of melt pond onset was consistent with other studies (Webster et al., 2015; Niehaus et al., 2023).*

We propose to revise this line as: "The timing of melt pond onset was consistent with other studies. For example, Webster et al.(2015) found that the first-year ice near the 2011 Applied Physics Laboratory Ice Station (APLIS) site in the Beaufort and Chukchi seas experienced pond onset near June 17. Similarly, Niehaus et al.(2023) showed that the melt pond onset occurred around mid-June to early-July during their Arctic-wide study between 2017 and 2021."

Line 203: the large difference could be also due to lack of data and not capturing the maximum MPF date

Response:26 We thank the reviewer for pointing this out. We agree with the reviewer, and we propose to add this line, "The difference in maximum MPF could also be attributed to the unavailability of data around maximum MPF date," to line 204.

Line 210: populated by mix of ice types in which year?

Response: 27 This is true for all four years analyzed (2018-2021). We will rephrase it in the revised manuscript.

Line 210: show the image during the seasonal maximum of 2108MPF and see if they are in fact MYI floes.

Response:28 Please see fig. 4A in the response to Reviewer 2. We will add this figure to the supplement. Further, we will use the EASE-grid Sea Ice Age data product from NSIDC and the guide for understanding and identifying old ice in summer from the National Research Council, Canada, to determine the age of ice floes.

Guide for understanding and identifying old ice in summer - NRC Publications Archive: https://nrc-publications.canada.ca/eng/view/object/?id=990fa669-b87a-4c97-bda5-7aa80ee947a4, last access: 12 February 2025.

Response:29 We follow the definition used by Perovich et al. (2002). By spatial variability, they mean the variation in melt pond coverage between and within ice floes. For example, they noted that the aerial imagery acquired during the 1998 Surface Heat Budget of the Arctic Ocean (SHEBA) field experiment exhibited considerable variable melt pond fraction between photographs captured along their survey line on the same day. Further, they also noted that even for the same ice floes, some portions had significant ponding while the others had very few, if any, melt ponds.

We will describe spatial variability as the "variation in melt pond coverage between and within ice floes" in the revised manuscript.

Perovich, D. K., Tucker III, W. B., and Ligett, K. A.: Aerial observations of the evolution of ice surface conditions during summer, Journal of Geophysical Research: Oceans, 107, SHE 24-1-SHE 24-14, https://doi.org/10.1029/2000JC000449, 2002.

Line 211: and what is the reason for "spatial variability of melt ponds"- does Perovich 2002 give an answer?

Response:30 No, they could not provide a reason for this variability. However, they propose integrating melt pond and ice properties derived from surface-based studies to help address this question.

Line 215: is the Neihaus data public? Could you look just specifically at their region to examine the differences?

Response:31 The S2 data used in their study is available to the public through the Copernicus Open Access Hub of the European Space Agency. Comparing their study location with EASE-grid Sea Ice Age data product shows that their study area in the Canadian Arctic (the closest to our study location) is predominantly populated by multi-year ice. Further, their study location includes regions of landfast ice, unlike our study area.

https://scihub.copernicus.eu/dhus/#/home
Niehaus, H., Spreen, G., Birnbaum, G., Istomina, L., Jäkel, E., Linhardt, F., Neckel, N., Fuchs, N., Nicolaus, M., Sperzel, T., Tao, R., Webster, M., and Wright, N.: Sea Ice Melt Pond Fraction Derived From Sentinel-2 Data: Along the MOSAiC Drift and Arctic-Wide, Geophysical Research Letters, 50, e2022GL102102, https://doi.org/10.1029/2022GL102102, 2023.

Line 215: if they only provide Arctic wide estimates of MPF it does not make sense to compare your results with theirs at all- of course the melt timing is different.

Response:32 We agree with the reviewer. Their time series includes Arctic-wide S2 imagery from different years and ice conditions (such as pack ice and landfast ice). (Also, please see response 31). Therefore, we propose to add this line: "Further, their time-series includes Arctic-wide S2 imagery from different years and ice conditions (such as pack ice and landfast ice). Therefore, we note that a direct comparison between our time-series and that of Niehaus et al. (2023) is not feasible" to line 221.

Line 225: by "smaller ponds < 200 m^2" doesn't that mean just one single Sentinel-2 pixel? Do you have a limit on the number of pixels required to call the object a pond? This is common of other studies- see Buckley et al 2023 and Perovich et al 2002 papers. If you eliminate small ponds which may be erroneous, you will probably have a totally different distribution to discuss.
Perovich, D. K., W. B. Tucker III, and K. A. Ligett, Aerial observations of the evolution of ice surface conditions during summer, J. Geophys. Res., 107(C10), 8048, doi:10.1029/2000JC000449, 2002.
Buckley, E. M., Farrell, S. L., Herzfeld, U. C., Webster, M. A., Trantow, T., Baney, O. N., ... & Lawson, M. (2023). Observing the evolution of summer melt on multiyear sea ice with ICESat-2 and Sentinel-2. The Cryosphere, 17(9), 3695-3719.

Response:33 We thank the reviewer for pointing this out. We agree with the reviewer that some of the smaller melt ponds may be erroneous due to the limitations of the spatial resolution of S2 imagery. Therefore, we have established a minimum threshold in order to analyze the melt pond size distribution as suggested by the reviewer. To establish the threshold, we used the S2 images acquired on June 27, 2020, and July 11, 2022. We binned the melt ponds into five categories based on the number of pixels in individual melt ponds. For example, Category 1 contains the melt ponds covered by one S2 pixel, Category 2 contains the melt ponds covered by two S2 pixels, and so on up to Category 5, which contains the melt ponds covered by five S2 pixels. Then, we selected 100 random melt ponds from each category and compared them with coincident WV images to determine the accuracy of melt pond classification in each category. Our analysis showed that the melt ponds in Category 4 and above had an accuracy of >95% (Table A). Therefore, we have determined the threshold to be four S2 pixels (400 m2), and we will modify sections 4.3 (figure 4) and 4.6 (figure 6) based on this threshold. Note that this threshold is in close agreement with Freitas et al. (2019), who found that S2 imagery may be suitable for monitoring thermokarst lakes and ponds larger than 350 m2.
Freitas, P., Vieira, G., Canário, J., Folhas, D., and Vincent, W. F.: Identification of a Threshold Minimum Area for Reflectance Retrieval from Thermokarst Lakes and Ponds Using Full-Pixel Data from Sentinel-2, Remote Sensing, 11, 657, https://doi.org/10.3390/rs11060657, 2019.

| Category | June 27, 2020 | | July 11, 2022 | |
| --- | --- | --- | --- | --- |
| | True Positives | False Positives | True Positives | False Positives |
| Category 1 | 98 | 2 | 87 | 13 |

| Category 2 | 94 | 6 | 92 | 8 |
| Category 3 | 98 | 2 | 92 | 8 |
| Category 4 | 99 | 1 | 96 | 4 |
| Category 5 | 99 | 1 | 95 | 5 |

Table A shows the accuracy of melt pond classification.

Line 232: see also Buckley et al., 2023 for melt pond evolution for comparison especially because it also uses Sentinel-2 data for MPF.

Response:34 Thank you. We will modify the section as mentioned in response 33 and will discuss Buckley et al., 2023.

Figure 4: I think this could be better represented with month and day on the y axis and then the data plotted across the season with each year as a different color- that would be easier to visualize the patterns and how they line up year to year.

Response:35 We thank the reviewer for their suggestion. We will modify the figure accordingly in the revised manuscript.

Line 239: was the performance the lowest or the presence of errors and misclassifications lowest? How can you know? Are these percentages a percentage of the misclassifications being removed? This is a confusing discussion, please clarify.

Response:36 Reviewer 2 has also found this section to be confusing. Therefore, we will put the numbers in a table and clarify the text as suggested by the reviewers.

Line 260: It would be beneficial to establish a lower limit for the detectable melt pond size. Given that a pond in the 100 m. to 200 m. size range corresponds to just a single Sentinel-2 pixel, the discussion around melt pond size distribution within this category lacks robustness. I would be particularly interested in seeing the distribution of ponds starting with a minimum size equivalent to 10 Sentinel-2 pixels. Additionally, it would be valuable to include a dedicated section discussing the limitations of the 10 m resolution, specifically addressing how many ponds and the total pond area may be missed compared to higher-resolution imagery, such as that from WV.

Response:37 Please see response: 33. We will also add a dedicated section to address the limitations and misclassifications, as mentioned in response:22.

Line 281: There is the stat I was looking for! Maybe include this earlier where noted.

Response:38 We will include this in the limitations section (please see response:22)

Line 299: The conclusions of the study feel overly optimistic and would benefit from a more balanced discussion of the methodology's limitations. Specifically, the following points should be addressed:
• Limitations of Sentinel-2 coverage: Sentinel-2 imagery is restricted to areas within a certain distance from the coastline, which limits its applicability in more remote regions. Additionally, temporal limitations should be discussed—how frequently are these specific locations sampled, and does this sampling frequency align with the temporal dynamics of melt pond evolution?
• Impact of cloud contamination: A detailed assessment of the extent to which cloud cover renders data unusable would provide valuable context. Quantifying the proportion of affected data could help clarify the reliability of the results.
• Manual aspects of the algorithm: The methodology currently involves manual corrections and decisions, such as selecting the optimal morphological operation. If this approach were scaled up, manual intervention would not be feasible, necessitating a discussion on how these steps could be automated or standardized.

Response: 39 We agree with the reviewer that the manuscript would benefit from discussing the limitations in the conclusion section. Therefore, we will modify the conclusion to address the reviewer's points on S2 coverage, temporal limitations, cloud cover, and the manual adjustment used in the algorithm.